# Extracellular Vesicle-Mediated Mitochondrial Reprogramming in Cancer

**DOI:** 10.3390/cancers14081865

**Published:** 2022-04-07

**Authors:** Roger Carles-Fontana, Nigel Heaton, Elena Palma, Shirin E. Khorsandi

**Affiliations:** 1The Roger Williams Institute of Hepatology, Foundation for Liver Research, London SE5 9NT, UK; roger.carles_fontana@kcl.ac.uk; 2Faculty of Life Sciences and Medicine, King’s College London, London WC2R 2LS, UK; nigel.heaton@nhs.net; 3Institute of Liver Studies, King’s College Hospital, NHS Foundation, London SE5 9RS, UK

**Keywords:** tumor-derived EVs (TEVs), miRNA, mitochondrial dynamics, metabolism, tumor microenvironment (TME)

## Abstract

**Simple Summary:**

Mitochondria are important organelles involved in several key cellular processes including energy production and cell death regulation. For this reason, it is unsurprising that mitochondrial function and structure are altered in several pathological states including cancer. Cancer cells present variate strategies to generate sufficient energy to sustain their high proliferation rates. These adaptative strategies can be mediated by extracellular signals such as extracellular vesicles. These vesicles can alter recipient cellular behavior by delivering their molecular cargo. This review explores the different EV-mediated mitochondrial reprogramming mechanisms supporting cancer survival and progression.

**Abstract:**

Altered metabolism is a defining hallmark of cancer. Metabolic adaptations are often linked to a reprogramming of the mitochondria due to the importance of these organelles in energy production and biosynthesis. Cancer cells present heterogeneous metabolic phenotypes that can be modulated by signals originating from the tumor microenvironment. Extracellular vesicles (EVs) are recognized as key players in intercellular communications and mediate many of the hallmarks of cancer via the delivery of their diverse biological cargo molecules. Firstly, this review introduces the most characteristic changes that the EV-biogenesis machinery and mitochondria undergo in the context of cancer. Then, it focuses on the EV-driven processes which alter mitochondrial structure, composition, and function to provide a survival advantage to cancer cells in the context of the hallmarks of cancers, such as altered metabolic strategies, migration and invasiveness, immune surveillance escape, and evasion of apoptosis. Finally, it explores the as yet untapped potential of targeting mitochondria using EVs as delivery vectors as a promising cancer therapeutic strategy.

## 1. Introduction

One of the established hallmarks of cancer is the metabolic switch cancer cells undertake to satisfy energy demands and provide biomolecules to support heightened proliferation and adapt to a microenvironment that is often scarce in oxygen and nutrients [1]. Metabolic dysregulation suggests that the reprogramming of mitochondria is involved due to the key role of this organelle in the production of energy and the metabolic intermediates necessary for biosynthesis.

The cancer metabolic phenotype is more dependent on external cues from the tumor microenvironment (TME) than originally thought. The TME is a complex network that includes tissue stromal cells, immune cells, vessels, extracellular matrix proteins, adipocytes, and soluble factors. The TME is crucial for supporting cancer cells throughout all their developmental stages and is modulated by intercellular communication [2]. Functional interactions between cancer cells and non-malignant cells in the TME are mediated by several mechanisms including extracellular vesicles (EVs).

EVs are small vesicles generated by all cell types, including cancer cells, that contain a wide range of bioactive molecules that can be transferred to recipient cells, activating or silencing signaling pathways and therefore modulating cellular behavior. Tumor-derived EVs (TEVs) are acknowledged to be involved in regulating many of the hallmarks of cancer such as escaping immune surveillance, promoting angiogenesis, evading apoptosis, and inducing a metabolic switch [3].

Given the central role that mitochondria play in many of these processes, this review focuses on the EV-mediated mechanisms used by cancer cells to reprogram the mitochondria of recipient cells to support their growth. We also highlight the bidirectional communication that occurs between cancer and healthy cells and how this process is critical for supporting cancer survival. This review then concludes by assessing the potential of EV-mediated mitochondrial reprogramming as a therapeutic strategy for cancer treatment.

## 2. Extracellular Vesicle Release and Cargo in Cancer

### 2.1. Overview

EVs are membranous vesicles delimited by a lipid bilayer without replicative capacity, with sizes ranging from 30 to 1000 nm that are generated by all cells and have been found in multiple biofluids [4]. They differ according to their biogenesis, size, cargo, and cellular pathophysiological state. The main EV types are those derived from the outward shedding of the plasma membrane, called microvesicles or ectosomes, and those derived from the endocytic pathway, named exosomes. However, due to the difficulty in determining the pathway of vesicle biogenesis in experimental settings, the International Society of Extracellular Vesicles (ISEV) recently recommended a nomenclature based on particle size (small EVs (<200 nm) and big EVs (>200 nm)) [4]. Nonetheless, new technologies are in development in order to achieve selective isolation of EV subtypes. Specific isolation and characterization of EV subtypes will be key to a better understanding of their particular functions as well as for their application in clinical practice [5].

Exosome biogenesis starts with the formation of the early endosome by the inward budding of the cellular membrane. The limiting membrane buds inwards, creating intraluminal vesicles (ILV) within the late endosome or multivesicular body (MVB). Finally, the MVB contents can be degraded by fusing with lysosomes or secreted by the merging of its membrane with the cellular bilayer, thereby releasing the ILVs to the extracellular milieu in the form of exosomes (Figure 1) [6,7]. Exosome biogenesis is mainly regulated by the endosomal sorting complexes required for transport (ESCRT) machinery. ESCRT-independent mechanisms also exist including those mediated by ceramides and tetraspanin proteins [8]. EVs play an important role in intercellular communication, as they can be taken up by surrounding cells or enter the systemic circulation and target distant tissues. The cargo of EVs includes proteins, metabolites, lipids, and nucleic acids (mRNA, non-coding RNA (ncRNA), DNA, mitochondrial DNA (mtDNA)). The release of this cargo inside recipient cells can modulate cellular behavior by affecting gene and protein expression, and nutrient abundance, eventually impacting cell homeostasis. This stands true in both physiological and pathological situations such as cancer. The biological importance of EVs in cancer is supported by the observations that there is both an increased release of EVs, and an alteration in their cargo [9,10].

### 2.2. Alterations in EV Biogenesis and Release

Cancer-enhanced EV release suggests a dysregulation in the components of the EV biogenesis machinery as a survival strategy. In support of this hypothesis, ESCRT proteins have been found to be upregulated in some cancer types [11,12,13] as well as other molecules involved in exosome biogenesis and release, such as heparanase [14,15] and syntenin [16]. Moreover, several oncogenic signaling pathways [reviewed in McAndrews et al. [17]] have been implicated in the upregulation of exosome biogenesis, however, further exploration of the underlying mechanisms is required. Extrinsic factors, such as environmental stressors, can also influence EV biogenesis. Cells subjected to stress conditions such as hypoxia, nutrient deprivation [18,19], mechanical strain [20], pH or with dysregulated protein synthesis [17] release more EVs as a compensatory mechanism.

### 2.3. Alteration in EV Cargo

The abundance of EV cargo molecules does not always reflect the intracellular content of their parental cell, indicating that these molecules are not passively loaded into EVs but through specific sorting systems that enrich for different cargo components (reviewed in [21,22]). Notably, EVs are highly enriched in small ncRNA such as microRNA (miRNA), which are increasingly recognized as playing an important role in gene regulation. The contribution of EV-derived miRNA to the hallmarks of cancer has been confirmed by many studies [23]. miRNAs typically bind to the 3′ untranslated region (UTR) of mRNA for translational repression. Interestingly, some miRNA species can also translocate into mitochondria and target mtDNA-derived mRNA [24] to alter the expression of the mitochondrial genome thereby modulating mitochondrial function [25]. The miRNA contained in EVs are enriched in specific short nucleotide motifs which act as loading signals, being recognized by RNA-binding proteins that interact with components of the ESCRT pathway [26,27,28]. Most of these RNA-binding proteins have been found to be dysregulated in cancer and are associated with poor clinical outcomes [29]. However, the mechanisms that these RNA-binding proteins use to interact with the endosomal machinery and thereby deliver the miRNA into the MVB need further study. The other main EV biomolecules that cancer can use to alter EV-recipient cell behavior are proteins [30]. These proteins can be sorted into EVs through post-translational modifications (PTMs) such as ubiquitination, a process that is often dysregulated in cancer [31]. These PTMs then act as loading signals to be recognized by the ESCRT pathway complexes [21].

EV nucleic acid and protein cargos have been extensively investigated for their potential role as biomarkers in cancer. However, lipids have also been found to be enriched in EVs, though their EV loading mechanisms remain largely unknown [32,33]. Most of these lipids are fatty acids, which can only be metabolized in mitochondria and can support cancer survival.

Despite lacking a comprehensive understanding of the cancer-specific mechanisms regulating EV biogenesis and cargo sorting, the alterations in these pathways, together with many functional studies, highlight the importance of EV biology in cancer progression.

## 3. Mitochondria in Cancer

Mitochondria are essential organelles that play a central role in the cellular metabolic axis, generating most of the energy necessary for cell function, as well as participate in other key cell events, such as regulating apoptosis and lipid homeostasis (Figure 2A). Therefore, it is no surprise that mitochondrial dysregulation is involved in the pathogenesis and progression of very diverse diseases, from muscular dystrophy [34] to alcohol-related liver disease [35,36] and cancer [37,38].

### 3.1. Mitochondrial Metabolism

Tumor cells undergo a series of metabolic changes that allow them to sustain an increased proliferation and to survive environmental stressors. Cancer metabolic adaptative strategies are heterogeneous, but many of them are connected to a reprogramming of mitochondria [39]. One of the most well-known and common alterations is the Warburg effect, namely a substantial increase in the glycolytic rate independent of the oxygen concentration. Despite the prevalence of glycolysis for energy production, mitochondrial oxidative phosphorylation (OXPHOS) is still required to support tumor formation, progression, and metastasis [38,40,41,42,43], highlighting the fundamental role that mitochondrial machinery plays in cancer biology.

In addition, the adaptation of cancer cells to alternative metabolite use can also be mediated by a mitochondrial reprogramming [44]. Interestingly, EVs have been found to modulate glucose metabolism in the context of cancer through a variety of cargo molecules including miRNA [45], long non-coding RNA (lncRNA) [46], integrins [47], enzymes and metabolites [48]. Fatty acid metabolism, which only occurs within the mitochondrial matrix, is also altered in cancer. Changes in fatty acid oxidation (FAO) are instrumental in cancer progression, metastasis, drug resistance and cell growth [49,50]. Again, EVs are implicated in lipid metabolism [51] and in the FAO changes of cancer, as they contain both mitochondrial FAO enzymes and substrates that are transferred to recipient cells [52,53,54,55,56]. In addition, glutamine, another mitochondrial metabolic substrate, is important for cancer cell growth [57] as both a source for tricarboxylic acid (TCA) cycle intermediates [58,59] as well as an antioxidant [60,61]. Increased levels of glutamine in the TME have been found to promote mitochondrial biogenesis [62]. Intriguingly, the cellular stress induced by glutamine deprivation leads to the release of EVs with a different formation mechanism containing an altered protumorigenic cargo [63].

### 3.2. Mitochondrial Dynamics

Effective mitochondrial functionality is tightly correlated with their number, size, mass, internal architecture, and intracellular location. Several processes are in place to regulate these parameters including fission, fusion, mitophagy, and intracellular trafficking which are collectively known as mitochondrial dynamics (Figure 2C). These events are important in defining cellular bioenergetics in health, but they can be perturbed in disease and promote tumorigenesis or tumor progression [64,65].

The alteration in the balance of mitochondrial fusion/fission events affects multiple aspects of tumor biology such as cell migration and proliferation, metabolism, cellular stemness and apoptosis [66,67]. Mitochondrial fusion and fission are mediated by several GTPases in the outer mitochondrial membrane such as dynamin-related protein 1 (Drp-1) (which promotes fission) and mitofusin 1 and 2 (Mfn-1/2) (which promote fusion). A typical change occurring in cancer is a higher expression and activity of Drp1, with the resulting increase in mitochondrial fission having been found to enhance the metastatic capacity and progression of breast [68,69] lung [70] and brain cancer [71]. In contrast, Mfn-2 has been found to inhibit the invasion and growth of lung [70] and breast [72] cancer. Moreover, low levels of Mfn-2 expression correlate with a poorer prognosis in breast cancer patients and low Mfn-1 is associated with an increased metastatic capacity in hepatocellular carcinoma (HCC) [65].

Mitochondrial fusion/fission processes are closely coordinated with the mitophagy machinery, which is responsible for the degradation and recycling of dysfunctional mitochondria. Mitophagy can be both tumor suppressive or protumorigenic depending on the tumor type, stage, or metabolic phenotype of the cancer [73]. In accordance with this dual role, mitophagy-related proteins have been found to be dysregulated in a number of cancers, being up or downregulated depending on cancer type [74]. Very recently, a functional connection has been found between EV secretion and mitophagy in melanoma through CD9, a member of the tetraspanin family. The inhibition of CD9 leads to mitochondrial malfunction that is compensated for by an increase in mitochondrial mass through a reduction in mitophagy [75].

Reported internal structural changes occurring in the mitochondria of cancer cells are related to mitochondrial cristae shape and density. Both are important as they affect the production of ATP by influencing function, stability and organization of the respiratory enzymes of the electron transport chain (ETC) into supercomplexes [76], thereby impacting all cellular processes that rely on mitochondrial respiration [77]. Further biological relevance of cristae architecture occurs when the cristae structure loosens allowing the release of proapoptotic factors [78]. The cristae structure has been found to be altered in cancer [79] and this has been attributed to changes in the composition of the mitochondrial contact site and cristae organizing system (MICOS) subunits that can vary according to cancer type [80].

### 3.3. Mitochondrial-Mediated Apoptosis

The mitochondrial dynamics machinery is also involved in the process of apoptosis or programmed cell death [81]. Resistance to apoptosis is one of the hallmarks of cancer and inducing apoptosis has been a leading cancer therapeutic approach for decades [82]. Apoptosis can either start through the activation of death receptors on the cell surface (extrinsic pathway) or through the involvement of mitochondria (intrinsic pathway) with molecular stressors increasing mitochondrial permeability leading to the release of cytochrome c. The intrinsic pathway is mainly regulated by the Bcl-2 family, through the balance of its proapoptotic (i.e., Bax, Bak) and antiapoptotic protein family members (i.e., Bcl-2, MCL-1, Bcl-XL) (Figure 2B) [83]. Cancer cells have been found to present an altered expression of Bcl-2 family members with EV cargo being found to modulate apoptosis by affecting Bcl-2 family members (reviewed in Section 4). Importantly, mitochondria are not passive actors in this process as mitochondrial architecture, lipid and protein composition are crucial in regulating apoptosis [81].

Taken together, these studies underline the active role of mitochondria in facilitating cancer survival through the adaptation of metabolic strategies, hijacking of mitochondrial dynamics, and suppression of the mitochondrial apoptotic pathways while emerging data is highlighting the role of EVs as mediators of these changes.

## 4. EV Modulation of Mitochondrial Processes in Cancer

Tumors are complex systems in constant communication with their microenvironment on which they rely for growth and survival. EVs, as intercellular communicators, are involved in several hallmarks of cancers, being active players in the remodeling of the TME and priming metastatic niches to support tumor survival, progression, and invasion. Although the importance of mitochondrial state and reprogramming in cancer progression has been established, the underlying mechanisms and metabolic phenotypes are incredibly varied, and knowledge is still lacking. Thus, the following section focuses on the role of EVs in mitochondrial reprogramming to support tumor survival by modulation of the TME in the context of the hallmarks of cancer (Table 1).

### 4.1. Dysregulating Cellular Energetics

#### 4.1.1. Metabolic Coupling with Cancer-Associated Fibroblasts (CAFs)

Among the different TME players, fibroblasts are key in metabolically supporting tumor progression [136]. Crosstalk between tumor cells and fibroblasts in the TME induces a phenotype change of the latter into cancer-associated fibroblasts (CAFs). CAFs display a hyperactive behavior facilitating tumor survival and migration [137]. During this transformation, fibroblasts switch to a glycolytic phenotype to provide cancer cells with energy-rich metabolic intermediates that fuel both mitochondrial ATP generation and biosynthesis [138]. This process, in which glycolytic CAFs support mitochondrial respiration in cancer cells, has been termed the reverse Warburg effect [139]. Reprogramming resulting from the crosstalk between cancer cells and local fibroblasts can occur through different mechanisms which can be mediated by TEVs (Figure 3A) [138].

One of these mechanisms is through the transfer of miRNAs affecting the expression of mitochondrial proteins. Melanoma TEVs have been found to contain miR-155 and miR-210 that lead to the reprogramming of stromal fibroblast metabolism by promoting glycolysis and inhibiting OXPHOS [87]. Similarly, miR-424, found in TEVs, downregulates the TCA enzyme IDH3a leading to the inhibition of OXPHOS through the upregulation of mitochondrial NDUFA4L2 [84,85]. Another mechanism of OXPHOS downregulation is through lung cancer TEVs transfer of miR-210 to CAFs [88]. Mechanistically, miR-210 directly downregulates ETC complex I subunit NDUFA4 and complex II subunit SDHD, resulting in mitochondrial dysfunction, as evidenced by an alteration in mitochondrial membrane potential. Interestingly, this miRNA also modifies mitochondria size and cristae organization [89]. In cancer associated with viral infection, it has been observed that TEVs contain viral proteins that, upon delivery to normal fibroblasts, induce their transformation into CAFs. This transformation is associated with a decrease in OXPHOS [90]. Moreover, further work on fibroblasts incubated with colorectal cancer (CRC) TEVs exhibited dysregulated expression of a number of mitochondrial proteins such as upregulation of a complex V subunit (ATP5H), TCA cycle enzyme IDH2 and the β-oxidation enzyme ECH1, as well as downregulation of mitochondrial protein translation factor TUFM and detoxifying enzyme ALDH2 [86].

In turn, CAF-derived EVs (CAF-EVs) also participate in cancer’s metabolic switch. For instance, breast cancer CAF-EVs transfer the lncRNA SNHG3 that inhibits OXPHOS and increases glycolysis, driving an increased tumor cell proliferation [91]. Similarly, Zhao and colleagues showed that CAF-EVs were able to inhibit OXPHOS through the transfer of miRNAs such as miR-22, let7a and miR-125b. Moreover, CAF-EVs can modulate mitochondrial metabolism by the transfer of lipids, amino-acids, and de novo complete TCA metabolites. Of note, the most abundant miRNAs found in CAF-EVs targeted OXPHOS genes with downregulation in the expression of subunits in complexes III and IV of the ETC [92]. The same group later demonstrated that CAF-EV-derived metabolites were able to support nutrient-deprived pancreatic cancer cell metabolism by modulating the TCA cycle [93]. Moreover, CAF-EVs delivered miR-92a to CRC [94]. Inhibition of this miRNA has been found to enhance oxygen consumption as well as increase the expression of mitochondrial proteins in adipocytes [95], suggesting a mitochondrial effect in cancer cells. However, fibroblasts metabolically reprogrammed by TEVs can also lead to increased OXPHOS, mitochondrial activity and number through the supply of energy-rich metabolites to cancer cells [90]. Consequently, the observed capacity of CAF-EVs to support the cancer cell TCA cycle while generally inhibiting OXPHOS, suggests a repurposing of mitochondria from an energy “powerhouse” to a biosynthetic “factory”.

EV-mediated mitochondrial reprogramming is bidirectional, the direction and nature of this reprogramming being dependent on cancer type and pathological stage. Supporting evidence for the changing role of EVs in the natural history of cancer comes from early and late-stage colorectal cancer TEVs being able to induce different phenotypes in fibroblasts, from protumorigenic to prometastatic phenotypes [86]. These observed differences in fibroblast function and phenotype suggest that the cargo of the TEVs can be adjusted to satisfy the varying needs of cancer cells throughout their developmental process.

#### 4.1.2. Metabolic Coupling with Adipocytes

Adipose tissue, composed mainly of adipocytes, has been found to promote tumor progression [140]. The main role of adipocytes is to maintain a physiological energy balance by storing fat and releasing it according to energy demands. Adipocyte function depends on its differentiation status which is linked to mitochondrial changes. Typically, differentiation of white adipocytes into brown/beige adipocytes is characterized by an increase in mitochondrial number and activity, as well as an increase in the expression of mitochondrial uncoupling protein (UCP1) that uncouples respiration from ATP synthesis to generate heat and stimulate lipolysis [141]. This phenotype change leads to the release of metabolites that, upon uptake by cancer cells, promote tumor progression (Figure 3A) [97].

TEVs have been demonstrated to induce lipolysis and browning in adipocytes through the transfer of miRNAs such as miR-155, miR-144 and miR-126. Consequently, adipocytes release fatty acids, glutamine, pyruvate, lactate, and ketone bodies for cancer cell utilization [96,97,142]. In parallel, experimental work with adipocytes in melanoma and prostate cancer has shown that adipocytes secrete EVs (adEVs) that increase their malignant potential. This observed phenotypic change was related to metabolic reprogramming, characterized by an increase in fatty acid catabolism via FAO. This catabolic switch was also accompanied by an increase in mitochondrial number and density [98]. The same group also showed that adipocytes can influence tumor metabolism by EV transfer of mitochondrial FAO enzymes, OXPHOS subunits, mitochondrial ADP/ATP transporters, TCA cycle proteins, and fatty acids [55]. AdEVs have also been shown to transfer miR-23a/b to HCC cells [99] and, in the context of prostate cancer, these miRNAs have been found to target mitochondrial glutaminase [100].

In a similar way to CAFs, cancer cells and adipocytes develop a metabolic relationship, where TEVs induce a metabolic shift in adipocytes encouraging the formation and release of energy-rich metabolites to provide nutritional support to cancer cells, thereby promoting a metabolic remodeling favoring FAO for their progression and invasion. This experimental evidence supports the need for further studies to investigate EV-mediated molecular mechanisms by which adipocytes can support tumor growth metabolically.

#### 4.1.3. Metabolic Subjugation of Neighboring Cells

The EV-mediated metabolic manipulation of healthy cells in cancer is not just limited to fibroblasts and adipocytes. Kaposi’s sarcoma-associated herpes virus-derived EVs transfer miRNA to uninfected cells resulting in reduced mitochondrial biogenesis and respiration, with an induction of aerobic glycolysis. The reduction in respiration observed was related to a decrease in mitochondrial volume in EV-treated cells by 40%. Overall, mitochondrial activity was also reduced as reflected by a lower level of TCA cycle metabolites [101]. Unfortunately, the underlying mechanisms leading to this alteration in mitochondrial biogenesis were not explored in greater depth. However, similarly to what occurs with CAFs and adipocytes, the purpose of this change in mitochondrial behavior is to turn EV recipient cells into feeder cells, by the release and transfer of glycolytic end products to infected cells to fuel their TCA cycle for growth advantage.

### 4.2. Resisting Cell Death

Resistance to apoptosis is a hallmark of cancer and apoptosis is a biological process that can be regulated by mitochondrial events. Chemotherapy ultimately exerts its function by inducing apoptosis which makes the mitochondrial state an important determinant of tumor response to therapy [143]. Mitochondrial dysfunction is associated with tumor cell resistance to both chemotherapy and radiotherapy by impeding apoptosis [144,145], with cancer cells being able to modulate mitochondrial apoptotic priming through EVs to benefit their survival (Figure 3D).

#### 4.2.1. Modulation of the Bcl-2 Pathway

Some of cancer’s EV strategies to evade apoptosis do not necessarily target mitochondrial proteins, but act indirectly on other aspects of the apoptotic cascade, for example transferring proteins such as survivin, which blocks cell death by inhibiting caspases [146], or miR-21 that targets the tumor suppressor programmed cell death 4 (PDCD4) [147]. However, there is also evidence that TEVs are taken up by melanoma and bladder cancer cells inhibiting apoptosis and favoring cell proliferation by targeting mitochondria through the regulation of the Bax/Bcl-2 signaling pathway [102,103]. However, the molecular basis of these observations needs further investigation.

EV-mediated resistance to apoptosis can also occur between cancer cells and other cell types in the TME. For instance, glioma cells transfer EVs containing lncRNA-CCAT2 to endothelial cells in order to promote angiogenesis by decreasing apoptosis in these cells through Bcl-2 increase and suppression of Bax and Caspase-3 expression [105]. The same effect also occurs in the opposite direction, such as between mesenchymal stem cells (MSCs) and cancer cells. MSCs are important players in maintaining tissue homeostasis as they promote processes involved in tissue regeneration, angiogenesis, and cell survival, and their EVs are one way they exert their function. However, MSC function can be hijacked by cancer to enhance its own survival [148]. MSC-derived EVs (MSC-EVs) have been found to contain lncRNA LINC00461 and, on uptake by multiple myeloma cells, it relieved the inhibitory effect of miR-15a/miR-16 on Bcl-2, therefore suppressing apoptosis and promoting myeloma proliferation [106].

#### 4.2.2. Chemotherapy Resistance

The antiapoptotic effect of TEVs has a direct implication on determining the response of tumor cells to therapy. TEVs can spread resistance to chemotherapeutic agents by transferring ncRNAs such as lncRNA-SNHG14, lncRNA PART1 and miR-214, as well as proteins such as chloride intracellular channel 1 (CLIC1) and connexin 43 to chemosensitive cells by modulating components of the Bcl-2 family [107,108,109,110,112,113]. Although the development of chemoresistance caused by the modulation of the Bcl-2 family appears to be common and consistent across a number of cancer and drug types, many of the underlying molecular mechanisms remain unknown.

Besides acting on mitochondrial apoptotic pathways, TEVs have been shown to modulate chemoresistance via the metabolic regulation of drug-sensitive cells. For instance, in doxorubicin-resistant breast cancer cells, TEVs were able to affect the balance between glycolysis and OXPHOS in recipient cells via mechanisms involving heat shock protein-70 (Hsp70). After Hsp-70 delivery by TEVs, this protein translocates into mitochondria and induces mitochondrial damage through increasing reactive oxygen species (ROS) levels. Consequently, recipient cells with impaired mitochondrial respiration switch to an enhanced glycolytic activity [111].

EVs from other cells in the TME also enhance drug resistance. For instance, CAF-EVs transfer miR-92a-3p and miR-103a-3p to cancer cells inhibiting mitochondrial apoptosis [114,149]. In another study, the co-culture of MSCs with tumor cells activated the mitochondrial fission mediator Drp1, inducing an increased fragmentation of mitochondria in a T-cell lymphoblastic leukemia model. This alteration in mitochondrial fission dynamics was related to a decrease in mitochondrial ROS, a glycolytic switch and contributed to the development of chemoresistance [150]. Although the specific extracellular mechanism was not studied, EV capacity to promote mitochondrial fission has been previously demonstrated [55,127].

#### 4.2.3. Enhancing Cancer Survival and Chemotherapy Resistance through EV Transfer of Functional Mitochondria and Mitochondrial Components

Besides the intercellular transfer of molecules such as miRNAs that regulate the expression of mitochondrial components, several studies have also described the transfer of whole functional organelles. The transfer of whole mitochondria has been proven to be a protective strategy for a range of conditions such as lung or stroke-induced injury which require a metabolic rescue [151,152]. There are a number of different mechanisms for horizontal transfer of mitochondria such as cell fusion, gap junctions, tunnelling nanotubes and EVs [153,154]. In the context of cancer, this is another strategy undertaken to alter mitochondrial respiration.

MSCs have been found to transfer whole mitochondria via big MSC-EVs to recipient cells and rescue their metabolic function (see Section 5). However, the mechanisms that are beneficial for regenerative medicine (e.g., cells with a metabolic disease) can also be exploited by cancer cells to increase their mitochondrial number for ATP production to favor their progression [155]. Genetic aberrations and high proliferation in tumor cells increase the likelihood of mitochondrial dysfunction and instability. Therefore, receiving functional mitochondria or mitochondrial components could help restore or enhance mitochondrial function. In the context of chemotherapy, it has been shown that the horizontal transfer of mitochondria from MSC to leukemic cells led to an increase in OXPHOS and cancer cell survival advantage [156].

Further evidence for EV transfer of mitochondria has been seen in the exchange from astrocytes to glioma cells. This was found to improve the metabolism of the recipient cancer cell and contribute to chemotherapy resistance. Mechanistically, mitochondria-derived NAD+ metabolic enzymes increase the availability of NAD+ in recipient cancer cells, inducing the PARP-mediated DNA repair pathway [115]. In turn, tumor-activated stromal cells were also found to transfer mitochondria to glioblastoma cells through a number of different mechanisms, including EVs, with the co-culture of both cell types increasing glioblastoma proliferation and resistance to anticancer treatments [116].

In addition to whole mitochondria, the transfer of mitochondrial components such as mtDNA [157], membrane proteins, and active mitochondrial enzymes also occurs. Mitochondrial proteins can account for up to 10% of the total protein content of small EVs [158]. TEVs from melanoma, ovarian, and breast cancer have all been found to contain mitochondrial components that were not present in healthy controls [159]. Interestingly, a recent study showed that trastuzumab was able to modulate the mitochondrial protein cargo of breast cancer cells with most of the affected proteins being involved in mitochondrial membrane organization. It was also observed that trastuzumab reduced mitochondrial cristae numbers. Therefore, the authors hypothesized that damaged mitochondrial components are packed into EVs for disposal. However, the effect of these EVs and their mitochondrial cargo on potential recipient cells needs further investigation [160].

mtDNA is also selectively packaged into EVs as demonstrated by the higher copy numbers found in EVs in blood compared to cell-free plasma and whole blood [161]. Specifically, cancer cells have been found to release TEVs carrying mtDNA [162]. Moreover, mtDNA copy numbers change according to cancer stage and when compared to healthy controls suggesting a potential role in cancer progression [157,163]. Further evidence for EV mtDNA involvement in cancer biology is from their ability to restore OXPHOS and induce resistance to hormonal therapy in breast cancer cells [117]. Tumor cells that lack mtDNA exhibit delayed tumor growth and cells in the TME can transfer mtDNA to mtDNA-depleted tumor cells thereby re-establishing mitochondrial respiration and enhancing tumorigenic potential. Although the transfer process was not studied, EVs were proposed as a potential mechanism [164]. In the context of chronic alcohol exposure, mitochondrial aldehyde dehydrogenase (Aldh2)-deficient hepatocytes produce oxidized mtDNA, which can be delivered into neighboring HCC cells via EVs activating multiple oncogenic pathways which promote HCC [118]. These results highlight a biological role for EV mtDNA cargo in cancer behavior.

### 4.3. Avoiding Immune Destruction

A relevant hallmark of cancer is immune evasion as cancer cells have the capacity to alter both immune surveillance and its response. Tumors utilize different strategies to silence the immune response in order to create a tolerant TME [165,166]. Cancer cells are known to produce immunosuppressive EVs that contain bioactive molecules with immunomodulatory effects [167]. Recent reviews extensively cover the various mechanisms of TEV-mediated immune suppression [168,169]. However, those affecting mitochondrial function remained largely unaddressed. Potential TEV-mediated immune suppression mechanisms involving the reprogramming of immune cell mitochondria include (i) modulating metabolism to inactivate their tumor suppressor function and (ii) triggering mitochondria mediated apoptosis (Figure 3B). Interestingly, immunoregulatory effects can also be induced by the transfer of whole mitochondria via EVs [170].

#### 4.3.1. Macrophage Mitochondrial Reprogramming in Cancer

In recent years, the link between metabolic reprogramming and immune cell function has attracted increasing attention, giving rise to the field of immunometabolism. The metabolic profile of immune cells can define their phenotype and antitumor capabilities [171]. One of the TEV-mediated mechanisms promoting immune evasion is through shifting macrophages towards an anti-inflammatory M2 phenotype [119,172,173]. Macrophage M2 polarization is considered a key component for tumor progression and is linked to mitochondrial metabolism [172]. However, little attention has been paid to TEVs and their role in mediating immune escape through macrophage mitochondrial reprogramming [173,174,175,176,177]. Proinflammatory M1 macrophages have suppressed mitochondrial function and increased glycolysis to support the shift in their energy demands. In contrast, M2 macrophages present an increase in mitochondrial respiration, a preserved TCA cycle and an enhanced fatty acid metabolism [172]. TEVs can alter mitochondrial respiration in non-committed M0 macrophages and polarize them to the M2 phenotype, and their miRNA cargo has been hypothesized as the basis for this observation [119]. Supporting experimental evidence comes from hypoxic TEV inducing M2-like polarization in infiltrating macrophages by enhancing mitochondrial OXPHOS through the transfer of let-7a miRNA which suppresses the insulin mediated mTOR signaling pathway [120]. Many of the underlying mechanisms of macrophage M2 polarization in cancer remain unexplored. Despite this, adaptations of the ETC complexes are recognized to play an important role in defining macrophage immune response by contributing to the immune metabolic switch [178]. Given the importance of bioenergetics in macrophage polarization and the ability of TEVs to modulate cell metabolism, further research is needed on EV-induced mitochondrial reprogramming of macrophages to potentially provide insight into how cancer promotes its own growth and immune escape.

#### 4.3.2. EV-Related Immunosuppression of T Cells

Other cell types, such as T cells, play a significant role in the regulation of antitumor immune response [179], therefore becoming the target of immunomodulatory mechanisms aimed to reduce cancer cell killing. TEVs have been found to induce apoptosis of activated T cells [180,181] via modulation of the Bcl-2/Bax pathway [123]. TEVs are also responsible for driving antitumor CD8+ effector T cell apoptosis by the presence of FasL and MHC class I as cargo molecules that trigger caspase activation. This, in turn, initiates the release of cytochrome c from mitochondria, leading to a loss of mitochondrial membrane potential and subsequent DNA fragmentation [124,181]. This process is not only modulated by TEVs but also by myeloid-derived suppressor cells EVs (MDSC-EVs). These cells are found in the TME and are recognized for their role in tumor progression. MDSC-EVs contain protumorigenic factors, such as death receptor proteins Fas and TNF-1α, inherited from their parental cells that are capable of immunosuppression by promoting mitochondrial apoptosis in CD8+ T cells through enhanced ROS [125]. The levels of internal ROS in antigen-specific T cells determine cell fate through reciprocal modulation of FasL and antiapoptotic Bcl-2 [126]. In addition to death ligands, melanoma TEVs can eliminate CD4+ T cells by directly targeting mitochondrial apoptosis regulator Bcl-2 through miR-690 [122]. Beyond Bcl-2 family regulation, melanoma cells were also found to enrich their EVs with certain RNAs that, upon delivery, altered mitochondrial function of tumor infiltrating cytotoxic T lymphocytes [121].

Another strategy employed by cancer cells to induce immunosuppression is through EV-encapsulated mtDNA. Mitochondria are a key source of damage-associated molecular patterns (DAMPs) and mtDNA is a large contributing component of mitochondrial DAMPs [182]. As described earlier (Section 4.2.3), TEVs have been found to have a higher copy number of mtDNA compared to healthy controls [163]. Mitochondrial Lon protease (Lon) plays many roles in mitochondrial homeostasis including ROS regulation. Lon drives ROS-induced mtDNA damage which results in its translocation into the cytosol. In the context of cancer, cytosolic mtDNA can affect the immune checkpoint receptors (known as the gatekeepers of the immune response) by causing overexpression of PD-L1 and subsequent activation of immune inhibitory pathways. In parallel, Lon increases the production of TEVs enriched with mtDNA and PD-L1 which inhibit T-cell activation. These Lon-induced EVs further contribute to cancer immune escape by silencing CD9+ T-cell immunity by inducing production of IFN and IL-6 from macrophages [183]. A further link between mitochondria and checkpoint receptors has been shown in a recent study, where the activation of the T-cell surface receptor PD-1 has been associated with downregulation of Drp1 phosphorylation on Ser616 and the consequent reduction of mitochondrial fission in T cells. The effect on mitochondrial dynamics causes a crucial impairment of T-cell functionality and therefore significantly affects antitumor response [184].

### 4.4. Activating Invasion and Metastasis

#### 4.4.1. Increased Motility and Migration

Alterations in mitochondrial dynamics play different roles in the TME by both inducing and supporting malignant transformation, and these effects can be mediated by TEVs. More specifically, EV-mediated dysregulation of mitochondrial dynamics appears to have a direct impact on cancer motility and migration capabilities (Figure 3C). For instance, hypoxic breast cancer cells have been shown to release EVs containing integrin-linked kinase (ILK) that increase epithelial cell migration, a requirement for epithelial-to-mesenchymal transition (EMT). As a result, mitochondrial movement was stimulated by promoting their intracellular trafficking and accumulation in the cortical cytoskeleton of epithelial cells. This observation was mediated by the phosphorylation of Ser616 in mitochondrial fission protein Drp1 combined with the increased expression of mitochondrial fusion proteins Mfn-1 and Mfn-2 [127]. Wu and colleagues similarly showed that TEVs could induce malignant transformation in healthy cells. EV-treated cells were found to have a higher number of small-sized mitochondria. The authors proposed that this effect was related to the downregulation of Mfn-2 altering the balance of mitochondrial fusion/fission, but further research is needed to elucidate this process [128].

Conversely, EVs derived from cells within the TME can also affect cancer cell mitochondrial dynamics. Adipocytes have been found to induce a more aggressive phenotype and promote invasiveness of cancer cells [96]. This observation can be achieved by the adipocyte transfer of FA and FAO enzymes via EVs to enhance FAO in cancer cells. This, in turn, alters mitochondrial dynamics to drive increased tumor cell migration. Specifically, increased FAO induces mitochondrial fission and intracellular redistribution of mitochondria to cell protrusions to facilitate cancer cell migration. In parallel, observed mitochondrial size reduction was attributed to EV transfer of mitochondrial fission regulators FIS1 and OPA1, with mitochondrial fission events being identified as crucial for cell migration [55]. CAF-EVs have also been shown to transfer miR-106b to pancreatic cancer where it has been associated with chemotherapy resistance [129]. On a mechanistic level, this miRNA species has been found to induce mitochondrial dysfunction in skeletal muscle by targeting Mfn-2 [130] but more studies are required in the context of cancer.

#### 4.4.2. Intravasation (Trans-Endothelial Migration into Vessels)

After increased cell motility, the next step in the metastatic cascade is the intravasation of tumor cells into the vasculature. TEVs have the capacity to destroy vessel barriers to promote this aspect (Figure 3C). Breast cancer TEVs are able to destroy endothelial barriers by activating endothelial to mesenchymal transition [131], which can be driven by mitochondrial dysfunction [132] and is required for the preparation of the premetastatic niche. miR-34a, present in TEVs [185], can also induce breakdown of the blood–brain barrier (BBB) by inducing a reduction in endothelial OXPHOS, ATP production and cytochrome c, the latter being a downstream target of miR-34a [186]. Another miRNA delivered through EVs that can promote destruction of the BBB is miR-181c [133]. A target of miR-181c is COX1 [134], whose downregulation results in mitochondrial apoptosis [135]. These preliminary but interesting results encourage the need for further investigation that strengthens the knowledge in the link between TEV cargo and the loss of endothelial barrier integrity via manipulation of mitochondrial biology.

## 5. Nanoparticle-Based Therapeutic Strategies Targeting Mitochondria in Cancer

EVs are key players in intercellular communication through their ability to deliver biological cargo to recipient cells. Thanks to this property, EVs have been considered as vectors for the delivery of both therapeutic endogenous and exogenous cargo. These therapeutic molecules can be loaded into EVs through a number of different mechanisms such as electroporation, or sonication [187]. Synthetic nanoparticles mimicking EVs are also being developed for drug delivery. These synthetic nanoparticles present the advantage of higher standardization, reproducibility, and yield, as well as control over the cargo. However, EVs have the advantage of presenting minimal toxic and immune-related side effects compared to synthetic nanoparticles. A number of studies have explored the potential use of EVs as a cancer therapy by suppressing tumor proliferation, invasion or reversing chemoresistance [188,189]. Given the role of mitochondrial reprogramming in cancer survival through metabolic adaptations and apoptosis resistance, this chapter focuses on the use of EVs as a vehicle to deliver molecules specifically targeting this organelle as a potential cancer therapeutic strategy (Table 2).

### 5.1. Naturally Occurring EVs

Different cells produce EVs that contain biomolecules with the capacity to eradicate tumor cells through mechanisms that specifically target their mitochondria. For instance, MSC-EVs have a number of potential therapeutic properties without the limitations of their parental cells [204]. Specifically, the potential of stem cell-derived EVs for therapy through the restoration of normal mitochondrial function and reduction of mitochondrial regulated apoptosis has been demonstrated in a variety of conditions such as kidney injury [205,206,207], myocardial infarction [208], Huntington’s disease [209], acute respiratory distress syndrome [210,211] or amyotrophic lateral sclerosis (ALS) [212,213,214]. Unfortunately, cancer cells can take advantage of the therapeutic effects of MSC-EVs in restoring mitochondrial function (see Section 4). Consequently, the role of MSC-EVs in cancer biology is still under investigation as there is supporting evidence for both pro- and antitumor properties [215,216]. For instance, MSC-EVs can induce apoptosis in ovarian cancer [216], HCC [215], leukemia [217], and breast cancer [218] by modulating Bcl-2 family proteins. This effect can be enhanced by combining MSC-EVs with chemotherapeutic agents [219,220]. Although the underlying molecular mechanisms remain unexplored, this effect could be mediated by miRNA cargo as MSC-EV miRNA has been shown to modulate pathways involved in mitochondrial function [216,221]. Furthermore, this mitochondria-mediated therapeutic action is not limited to apoptosis. In the context of cancer cachexia, muscle stem cell derived-EVs can reverse mitochondrial injury through their protein cargo by improving maximal oxygen consumption rate and spare respiratory capacity [222,223], which is a parameter indicative of mitochondrial adaptability to stress conditions [224].

The role of immune cells in killing abnormal cells suggests that their EVs might also hold therapeutic potential. EVs isolated from immune cells have been found to suppress tumor growth [225]. For instance, natural killer (NK)-derived EVs (NK-EVs) have recognized cytotoxic effects and can kill cancer cells through a number of different mechanisms. NK-EVs contain proteins such as granzyme A (GzmA), granzyme B (GzmB) and granulysin (GNLY) all of which have been reported to induce mitochondrial damage [226]. In particular, GzmB induces apoptosis by activating BIG, a member of the proapoptotic Bcl-2 family [227]. NK-EVs have also been shown to induce mitochondria-mediated apoptosis in cancer cells by triggering the release of cytochrome c [226] and altering the expression of Bcl-2 family members [228].

Interestingly, TEVs have also been found to have antitumor properties that can be achieved by activating the mitochondrial apoptotic pathway [229], arresting ATP production [230], and enhancing mitochondrial fusion [69,231]. Unfortunately, the evidence for these effects across different cancer types and the responsible underlying molecular mechanisms is sparse. One example of this effect is the higher level of let-7a found in CRC EVs. This miRNA is able to suppress mitochondrial OXPHOS and, therefore, ATP synthesis through the Lin28a/SDHA signaling pathway, thereby slowing tumor progression [230]. Apart from human cell-derived EVs, EVs from bacteria present in the intestines have also been shown to have antitumor effects on hepatic cancer cells by increasing the Bax/Bcl-2 expression ratio [232].

### 5.2. Altered EVs

#### 5.2.1. EVs Enriched with Biological Cargo

The high EV efficiency in delivering their cargo to recipient cells has led to the development of several strategies to enrich EVs with therapeutic biomolecules such as miRNAs. Many of these miRNAs are found to be downregulated in cancer and can induce their antitumor effect by altering mitochondrial function (Table 2). For instance, miR-126-enriched EVs can inhibit cell growth by modulating mitochondrial metabolism [191,192]. EVs can also be loaded with different miRNAs that induce cell death via the intrinsic mitochondrial pathway by downregulating Bcl-2 family antiapoptotic proteins [190,193,194,195]. Bcl-2 downregulation can also be achieved by delivering alternative EV cargo biomolecules including silencing RNA (siRNA) and antisense oligonucleotides (ASOs) [196,197] or by depleting EVs of tumorigenic circular RNA (circRNA) [104]. Interestingly, targeting mitochondrial proteins are not the only therapeutic option, as the knock down of antisense non-coding mitochondrial RNA (ASncmtRNA) via EVs was also shown to reduce tumorigenesis [198].

#### 5.2.2. EVs Loaded with Exogenous Drugs

Several cancer therapy strategies have focused on sensitizing cells to mitochondrial apoptosis, for instance, via Bcl-2 inhibitors [233]. In order to deliver these compounds more efficiently and selectively, chemotherapeutic drugs can be packaged into EVs (Table 2). Examples of this approach have been shown with manganese carbonyl and doxorubicin, which selectively induced mitochondrial damage in breast and lung cancer cells [199,200]. In other studies, staphylococcal enterotoxin B-coated TEVs were able to induce apoptosis by targeting Bcl-2 family proteins in estrogen receptor-negative breast and pancreatic cancer cells [201,202]. Apart from TEVs, NK-EVs loaded with paclitaxel were also found to enhance their antitumor effects by inducing apoptosis through the regulation of the Bcl-2/Bax axis [203].

Unsurprisingly, most of the presented EV-mediated strategies focus on reprogramming the mitochondria by inducing apoptosis via the regulation of Bcl-2 family. However, the importance of metabolism in tumor progression has led to the development of clinical trials targeting mitochondrial metabolic pathways as cancer therapy [43]. Given the evidence on the capacity of EV cargo to modulate mitochondrial metabolism, innovative approaches should include cancer therapeutic strategies aimed at tailoring EVs to deliver molecules, such as miRNA, that target this pathway.

## 6. Conclusions

In the last decade, EVs have emerged as important intercellular communicators involved in the development and progression of a number of pathological conditions including cancer. Cancers exploit dysregulated metabolic pathways to adapt to stress conditions, resist cell death, and alter biosynthesis to sustain increased proliferation. All these events point to a reprogramming of the mitochondria. EVs, through their cargo molecules, can alter mitochondrial biology to give cancer advantages such as metabolic benefits, immune escape, enhanced invasion, and apoptosis escape. However, many of the underlying mechanisms have not been explored in depth and the limited evidence in some cases presents contradictions that highlight the complexity and heterogeneity of cancer mitochondrial phenotypes. For this reason, further studies are needed to better understand the EV-mediated mechanisms and signaling pathways that modulate mitochondria in the TME. Obtaining this knowledge, together with the EV capacity to efficiently transfer molecules to recipient cells, holds the potential to offer new cancer therapeutic avenues based on restoring mitochondrial function using EVs as delivery vectors.

## Figures and Tables

**Figure 1 cancers-14-01865-f001:**
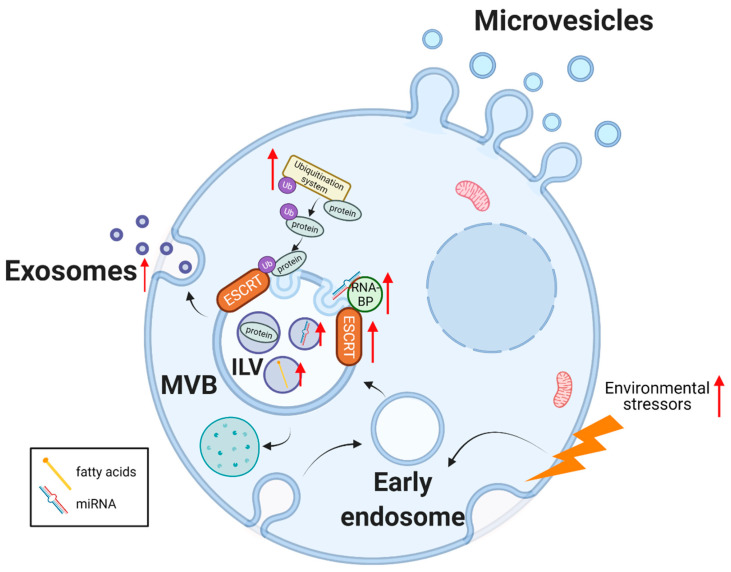
EV biogenesis, release, and cargo in cancer. Exosome biogenesis starts with the inward budding of the plasma membrane (early endosome). The limiting membrane buds inwards, generating intraluminal vesicles (ILV) contained in a late endosome or multivesicular body (MVB). The membrane of the MVB body fuses with the plasma membrane releasing the exosomes to the extracellular milieu. Several of the elements involved in exosome biogenesis and cargo sorting are upregulated in cancer (marked with a red arrow). Microvesicles or ectosomes are generated by the direct outward budding of the plasma membrane. Abbreviations used: ESCRT: endosomal sorting complexes required for transport. RNA-BP: RNA binding proteins. Ub: ubiquitin. miRNA: microRNA. Figure created with BioRender.com (accessed on 31 March 2022).

**Figure 2 cancers-14-01865-f002:**
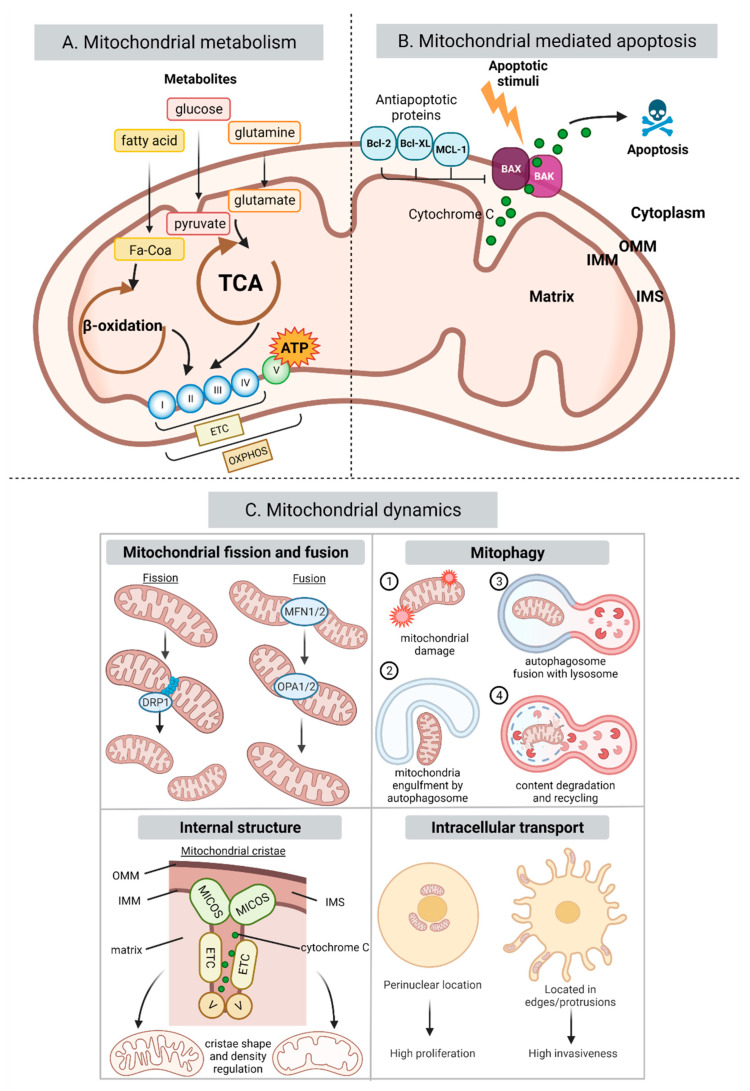
Mitochondrial processes affected in cancer. (**A**) Summarizes key metabolic processes in mitochondria. Metabolites such as glucose, fatty acids (FA) and glutamine are transformed into pyruvate, FA-CoA and glutamate and enter the TCA and β-oxidation cycles taking place in the mitochondrial matrix. These processes are used by the ETC to pump protons into the intermembrane space creating a chemical gradient used by complex V to generate ATP in a process called OXPHOS. (**B**) Summarizes the role of the Bcl-2 family proteins in the mitochondrial intrinsic pathway. Briefly, apoptotic stimuli lead to the formation of the pores in the OMM by Bax and Bak releasing cytochrome c into the cytoplasm which triggers the caspase cascade inducing apoptosis. The antiapoptotic members of the Bcl-2 family inhibit this process. (**C**) Summarizes the four main mitochondrial dynamics processes: mitochondrial fusion and fission, mitophagy, internal cristae architecture, and mitochondrial intracellular trafficking. In cancer, all these mitochondrial activities are dysregulated to support cancer energetic and biosynthetic demands, as well as survival. Abbreviations used: TCA: tricarboxylic acid. ATP: adenosine triphosphate. ETC: electron transport chain. OXPHOS: oxidative phosphorylation. I: complex I. II: complex II. III: complex III. IV: complex IV. V: complex V. OMM: outer mitochondrial membrane. IMM: inner mitochondrial membrane. IMS: intermembrane space. MICOS: mitochondrial contact site and cristae organizing system. Figure created with BioRender.com (accessed on 31 March 2022).

**Figure 3 cancers-14-01865-f003:**
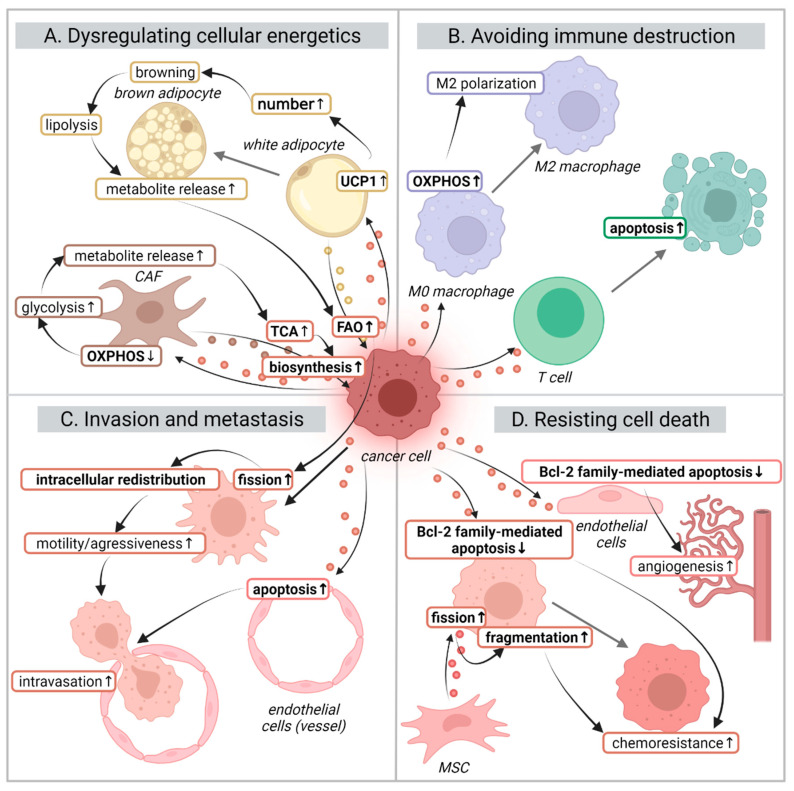
EV modulation of mitochondria and its effects in the context of hallmarks of cancer. Cancer cells generate tumor-derived EVs (TEVs) that alter the mitochondrial behavior of different cell types of the tumor microenvironment to support cancer progression, invasion, and survival. Through these effects, TEVs are mediators of several of the hallmarks of cancer such as: (**A**) dysregulation of cellular energetics: cancer cells can induce the release of metabolites from CAFs and adipocytes to be used for cancer cell biosynthesis and FAO; (**B**) avoiding immune destruction: TEVs can induce immunosuppression by inducing M2 polarization in macrophages or T cell apoptosis via the mitochondrial intrinsic pathway; (**C**) promoting cell motility and facilitating invasiveness: increased FAO leads to an intracellular trafficking of mitochondria towards edge protrusions facilitating their migration. At the same time, they can induce mitochondrial apoptosis in endothelial barrier cells to permit intravasation; (**D**) resisting cell death: alterations in Bcl-2 family members leads to inhibition of mitochondrial apoptosis resulting in chemotherapy resistance and angiogenesis. Abbreviations used: OXPHOS: oxidative phosphorylation. TCA: tricarboxylic acid cycle. FAO: fatty acid metabolism. UCP1: uncoupling protein 1. MSC: mesenchymal stem cell. Labels in bold represent mitochondria-related elements and processes. The colors of the EVs reflect those of the parental cell that produced them. The colors of the text borders reflect the cells where the indicated processes take place. Figure created with BioRender.com (accessed on 31 March 2022).

**Table 1 cancers-14-01865-t001:** EV-mediated modulation of mitochondrial processes in cancer. This table summarizes EV studies according to the hallmarks of cancer where a mitochondrial effect was observed in recipient cells. Abbreviations and symbols used: n.i.: not investigated. NA: not applicable. CRC: colorectal cancer. OXPHOS: oxidative phosphorylation. TCA: tricarboxylic acid. CI: complex I. CIII: complex III. CIV: complex IV. CV: complex V. CAF: cancer-associated fibroblast. LncRNA: long non-coding RNA. FA: fatty acid. FAO: fatty acid oxidation. HCC: hepatocellular carcinoma. MSC: mesenchymal stem cell. HNSCC: head and neck squamous cell carcinoma. NSCLC: non-small cell lung cancer. MDSC: myeloid-derived suppressor cell. Mito: mitochondria/l. ↑: increase. ↓ decrease. →: leading to.

EV Donor Cell	EV Recipient Cell	EV Cargo	ncRNA Target	Mitochondrial Effect	References
**Dysregulation of cellular energetics**					
Metabolic coupling with CAFs					
Prostate cancer cells	Fibroblasts	miR-424	n.i.	IDH3a↓, NDUFA4L2(CI)↑→OXPHOS↓	[84,85]
CRC cells	Fibroblasts	n.i.	n.i.	ATP5H(CV)↑, IDH2↑, ECH1↑, TUFM↓, ALSDH2↓	[86]
Melanoma cells	Fibroblasts	miR-155, miR-210	n.i.	OXPHOS↓	[87]
Lung cancer cells	Fibroblasts	miR-210	NDUFA4 (CI), SDHD (CII)	SDHD↓ → OXPHOS↓	[88,89]
Nasopharyngeal carcinoma cells	Fibroblasts	LMP1	NA	OXPHOS↓	[90]
CAFs	Breast cancer cells	lncRNA SNHG3	n.i.	OXPHOS↓	[91]
CAFs	Prostate cancer cells	miR-22, let7a, miR-125b and metabolites (amino acids, lipids, TCA intermediates)	n.i.	CYTB(CIII)↓, COXI (CIV) ↓→ OXPHOS↓	[92]
CAFs	Pancreatic cancer cells	miRNA, TCA metabolites	n.i.	TCA↑	[93]
CAFs	CRC cells	miR-92a	n.i.	OXPHOS↓	[94,95]
Metabolic coupling with adipocytes					
Breast cancer cells	Adipocytes	miR-155	PPARγ	UCP1↑	[96]
Breast cancer cells	Adipocytes	miR-144, miR-126	n.i.	UCP1↑, mito matrix density↑	[97]
Adipocytes	Melanoma, prostate tumor cells	FAO proteins (ECHA, HCDH), TCA proteins, OXPHOS proteins	NA	Mito number and density↑, FAO↑	[98]
Adipocytes	Melanoma cells	FAO enzymes (HCDH, ECHA, HCD2), OXPHOS subunits (NDUA6 and NDUAS2 (CI) and ATPG (CV)), mitochondrial ADP/ATP transporters (ADT1, ADT2 and ADT3), fatty acids	NA	FAO↑, mito activity↑, mito intracellular trafficking↑	[55]
Adipocytes	HCC cells	miR-23a/b	Mitochondrial glutaminase	Glutamine metabolism regulation	[99,100]
Metabolic subjugation of neighboring cells					
Kaposi’s sarcoma-associated herpes virus cells	Uninfected cells	miRNA	n.i.	Mito biogenesis↓, mito respiration↓	[101]
**Resistance to cell death**					
Modulation of the Bcl-2 pathway					
Melanoma cells	Melanoma cells	n.i.	n.i.	Bax↓, Bcl-2↑	[102]
Bladder cancer cells	Bladder cancer cells	n.i.	n.i.	Bax↓, Bcl-2↑	[103]
Acute myeloid leukemia cells	Acute myeloid leukemia cells	circ_0009910	miR-5195–3p	Bax↓, Bcl-2↑	[104]
Glioma cells	Endothelial cells	lncRNA-CCAT2	n.i.	Bax↓, Bcl-2↑	[105]
MSCs	Multiple myeloma cells	lncRNA LINC00461	miR-15a/16	Bcl-2↑	[106]
Chemotherapy resistance					
Trastuzamab-resistant breast cancer cells	Trastuzamab-sensitive breast cancer cells	lncRNA snhg14	n.i.	Bax↓, Bcl-2↑	[107]
Gefitinib-resistant esophageal squamous cell carcinoma cells	Esophageal squamous cell carcinoma cells	lncRNA PART1	miR-129	Bax↓, Bcl-2↑	[108]
Gefitinib-resistant non-small cell lung cancer cells	Gefitinib-sensitive non-small cell lung cancer cells	miR-214	n.i.	Bax↓, Bcl-2↑	[109]
Vincristine-resistant gastric cancer cells	Vincristine-sensitive gastric cancer cells	CLIC1	NA	Bcl-2↑	[110]
Doxorubicin-resistant breast cancer cells	Doxorubicin-sensitive breast cancer cells	Hsp70	NA	mito damage↑→ OXPHOS↓	[111]
Temozolomide-resistant glioblastoma cells	Temozolomide-sensitive glioblastoma cells	Connexin 43	NA	Bax↓, Bcl-2↑	[112]
Gefitinib-treated NSCLC cells	Untreated NSCLC cells	n.i.	n.i.	Bax↓, Bcl-2↑	[113]
CAFs	CRC cells	miR-92a-3p	n.i.	Bax↓	[114]
Enhancing survival and chemoresistance through transfer of mitochondrial components					
Astrocytes	Glioma cells	mitochondria	NA	Mito metabolism↑	[115]
Tumor-activated stromal cells	Glioblastoma cells	mitochondria	NA	OXPHOS↑	[116]
CAFs	Breast cancer cells	mtDNA	NA	OXPHOS↑	[117]
Aldh2-deficient hepatocytes	HCC cells	Oxidized mtDNA	NA	Bcl-2↑	[118]
**Escape from immune destruction**					
Macrophage mitochondrial reprogramming in cancer					
Lung tumor cells	M0 macrophages	n.i.	n.i.	CI↓→OXPHOS ↓	[119]
Hypoxia-induced TEVs	Infiltrating macrophages	let-7a	n.i.	OXPHOS↑	[120]
Immunosuppression of T cells					
Melanoma cells	Tumor infiltrating cytotoxic CD8+ T lymphocytes	n.i.	n.i.	OXPHOS↑	[121]
Melanoma cells	CD4+ T helper cells	miR-690	n.i.	Bcl-2↓, MCL-1↓, Bcl-xl↓	[122]
Renal carcinoma cells	Jurkat T Lymphocytes	FasL	NA	Bax↑, Bcl-2↓	[123]
HNSCC cells	CD8+ T cells	FasL	NA	Release of CytC, mito membrane potential↓, Bcl-2↓, Bcl-XL↓, Bax↑, Bim↑	[124]
MDSCs	CD8+ T cells	immuno-modulatory cytokines	NA	Mito ROS↑ → Bcl-2↓	[125,126]
**Activation of invasion and metastasis**					
Increased motility and migration					
Hypoxic breast cancer cells	Epithelial cells	n.i.	n.i.	Drp1 phosphorylation, Mfn-1↑, Mfn-2↑, mito intracellular movement↑	[127]
Bladder cancer cells	Urothelial cells	n.i.	n.i.	Mito size↓	[128]
Adipocytes	Melanoma cells	Fission regulators (FIS1, OPA1), FA, FAO enzymes, OXPHOS proteins	NA	fission↑, mito size↓, mito intracellular movement↑	[55]
CAFs	Pancreatic cancer cells	miR-106b	n.i.	Mfn-2↓	[129,130]
Intravasation					
Breast cancer cells	Liver sinusoidal endothelial cells	n.i.	n.i.	Mito disfunction	[131,132]
Brain metastatic cancer cells	Endothelial cells	miR-181c	COX1	Bax↑, Bcl-2↓	[133,134,135]

**Table 2 cancers-14-01865-t002:** Artificially modified EVs as delivery vectors for therapeutic cargo targeting mitochondria. This table summarizes the literature on the use of modified EVs as delivery vectors to obtain a therapeutic effect in the context of cancer by targeting the mitochondria. “EV source” refers to the cell type or biomaterial from which EVs were isolated. “Target cancer” refers to the cancer type investigated in the study. “Enrichment method” refers to the method by which EVs were enriched with the therapeutic molecule of interest. “(Bio)molecule” refers to the therapeutic molecule loaded into EVs. “Target” refers to the mitochondrial component targeted by the therapeutic biomolecule used. “Study type” classifies the studies into in vitro and in vivo depending on the models used. Abbreviations and symbols used: n.i.: not investigated. NA: not applicable. MSC: mesenchymal stem cell. HUVEC: human umbilical vein endothelial cells. ASO: antisense oligonucleotide. ASncmtRNA: antisense non-coding mitochondrial RNA. Mito: mitochondria/l. ↑: increase. ↓ decrease.

EV Source	Target Cancer	Enrichment Method	(Bio)molecule	Target	Effect on Recipient Cell Mitochondria	Study Type	References
**EVs enriched in biological cargo**							
MSCs	Breast	Cell transfection with lentivector	miR-34a	Bcl-2	Bcl-2↓	In vitro	[190]
HUVECs	Malignant mesothelioma	Cell transfection with transfection reagent	miR-126	IRS1	Affected mito metabolism, mito respiration↓	In vitro	[191,192]
Human embryonic kidney (HEK293) cells	Pancreatic	EV transfection via ultrasound	miR-34	Bcl-2	Bcl-2↓	In vitro and in vivo	[193]
Human bladder cancer (BIU-87) cells	Bladder	Cell transfection via viral vector (adenovirus)	miR-29c	n.i.	Bcl-2↓, MCL-1↓	In vitro	[194]
Breast cancer (MDA-MB-231 class) cells	Breast	Cell transfection	miR-205	n.i.	Bcl-2↓	In vitro	[195]
Bovine milk	Pancreatic and colorectal	EV transfection via ultrasound	siRNA	Bcl-2	Bcl-2↓	In vitro and in vivo	[196]
Acute myeloid leukemia (HL-60 and MOLM-13) cells	Acute myeloid leukemia	Cell transfection with lipofectamine	siRNA	circ_0009910	Bcl-2↓, Bax↑	In vitro	[104]
Hepatocellular carcinoma (HepG2) cells	Liver	EV transfection via coincubation and oscillation	ASO-G3139	Bcl-2	Bcl-2↓	In vitro	[197]
Breast cancer (MDA-MB-231) cells	Breast	Cell transfection with lipofectamine	ASO-1537S	ASncmtRNA	ASncmtRNA↓	In vitro and in vivo	[198]
**EVs loaded with exogenous drugs**							
Tumor cells	Metastatic breast	EV transfection with electroporation	Manganese carbonyl (MnCO)	NA	CO generation which induces mito toxicity	In vitro and in vivo	[199]
Non-small cell lung cancer (H1299) cells	Lung	EV coincubation with gold nanoparticles conjugated with doxorubicin	Doxorubicin	NA	Mito damage (perturbation of mito membrane potential), apoptosis via mito pathway↑	In vitro	[200]
Epithelial-like breast cancer (MDA MB-231) cell	Estrogen receptor negative breast	Protein anchorage	Staphylococcal enterotoxin B (SEB)	NA	Bax↑, Bak↑, Bcl-2↓	In vitro	[201]
Epithelial-like pancreatic cancer (MIA Paca-2)	Pancreatic	Protein anchorage	Staphylococcal enterotoxin B (SEB)	NA	Bax↑, Bak↑	In vitro	[202]
Natural Killer (NK-92)	Breast	EV transfection via electroporation	Paclitaxel	NA	Bax/Bcl-2↑	In vitro	[203]

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
