# Peer review of "Extracellular Vesicle-Mediated Mitochondrial Reprogramming in Cancer"

_cancers, 2022, doi:10.3390/cancers14081865_

Round 1

Reviewer 1 Report

This review manuscript by Carles-Fontana et al. untitled “Extracellular vesicle mediated mitochondrial reprogramming in Cancer” and submitted to Cancers journal is well written, very clear and relevant to the field. Here are my suggestions:

  • In the Introduction part, after:” The TME is crucial for supporting cancer cells throughout all their developmental stages and is modulated by intercellular communication.” a reference is lacking. Please add this paper which perfectly reflects the author’s thoughts: Cassim S, Pouyssegur J. Tumor Microenvironment: A Metabolic Player that Shapes the Immune Response. Int J Mol Sci. 2019 Dec 25;21(1):157. doi: 10.3390/ijms21010157. PMID: 31881671; PMCID: PMC6982275.
  • In the paragraph 3 named Mitochondria in Cancer, this citation should definitely be added: Cassim S, Vučetić M, Ždralević M, Pouyssegur J. Warburg and Beyond: The Power of Mitochondrial Metabolism to Collaborate or Replace Fermentative Glycolysis in Cancer. Cancers (Basel). 2020 Apr 30;12(5):1119. doi: 10.3390/cancers12051119. PMID: 32365833; PMCID: PMC7281550. It indeed absolutely reflects how mitochondria are altered/reprogrammed during cancer initiation and progression and is not old-dated – published in 2020.
  • In the paragraph 3.2 named Mitochondria metabolism, these 2 recent papers should definitely be added:
    • Martínez-Reyes I, Chandel NS. Cancer metabolism: looking forward. Nat Rev Cancer. 2021 Oct;21(10):669-680. doi: 10.1038/s41568-021-00378-6. Epub 2021 Jul 16. PMID: 34272515.
    • Vasan K, Werner M, Chandel NS. Mitochondrial Metabolism as a Target for Cancer Therapy. Cell Metab. 2020 Sep 1;32(3):341-352. doi: 10.1016/j.cmet.2020.06.019. Epub 2020 Jul 14. PMID: 32668195; PMCID: PMC7483781.
  • This title: “EV modulation of the hallmarks of cancer via mitochondrial processes” for paragraph number 4 should be modified because it is not clear at all.
  • In the paragraph dealing with CAFs, recent papers from the group of F. Mechta-Gigoriou should be added.
  • Some parts of the manuscript are too long in my opinion (especially at the end of the manuscript) and should therefore be shortened in order to keep the take-home message of this review clear, focused, and relevant for the readers. Otherwise, readers could be lost since a lot of information (even if they are of great interest) can be found in this work.

Author Response

Dear Reviewer 1,

Thank you very much for your time and effort in reviewing our manuscript and your constructive suggestions. We are pleased to hear you found it well written, clear, and relevant. Please see below our point-by-point response to your comments:

  1. In the Introduction part, after:” The TME is crucial for supporting cancer cells throughout all their developmental stages and is modulated by intercellular communication.” a reference is lacking. Please add this paper which perfectly reflects the author’s thoughts: Cassim S, Pouyssegur J. Tumor Microenvironment: A Metabolic Player that Shapes the Immune Response. Int J Mol Sci. 2019 Dec 25;21(1):157. doi: 10.3390/ijms21010157. PMID: 31881671; PMCID: PMC6982275. (page 1, line 43)

In the paragraph 3 named Mitochondria in Cancer, this citation should definitely be added: Cassim S, Vučetić M, Ždralević M, Pouyssegur J. Warburg and Beyond: The Power of Mitochondrial Metabolism to Collaborate or Replace Fermentative Glycolysis in Cancer. Cancers (Basel). 2020 Apr 30;12(5):1119. doi: 10.3390/cancers12051119. PMID: 32365833; PMCID: PMC7281550. It indeed absolutely reflects how mitochondria are altered/reprogrammed during cancer initiation and progression and is not old-dated – published in 2020. (page 5, line 148 and page 6, line 176)

In the paragraph 3.2 named Mitochondria metabolism, these 2 recent papers should definitely be added:

Martínez-Reyes I, Chandel NS. Cancer metabolism: looking forward. Nat Rev Cancer. 2021 Oct;21(10):669-680. doi: 10.1038/s41568-021-00378-6. Epub 2021 Jul 16. PMID: 34272515. (page 6, line 172)

Vasan K, Werner M, Chandel NS. Mitochondrial Metabolism as a Target for Cancer Therapy. Cell Metab. 2020 Sep 1;32(3):341-352. doi: 10.1016/j.cmet.2020.06.019. Epub 2020 Jul 14. PMID: 32668195; PMCID: PMC7483781. (page 6, line 176)

Many thanks for the suggestions. All the recommended references are very relevant, adding to the completeness of the review, and have been accordingly included in the revised manuscript (page and line can be found next ot each reference). In addition, we considered the recommended paper from Vasal et al. very interesting and appropriate to include also in Chapter 5 (page 22, line 724).

  1. This title: “EV modulation of the hallmarks of cancer via mitochondrial processes” for paragraph number 4 should be modified because it is not clear at all.

Thank you for your comment, we have now changed the title to “EV modulation of mitochondrial processes in cancer”.

  1. In the paragraph dealing with CAFs, recent papers from the group of F. Mechta-Gigoriou should be added.

We have included the reference “Tumor Cells and Cancer-Associated Fibroblasts: An Updated Metabolic Perspective”, 2021 by Gentric and Mechta-Grigoriou in the CAF paragraph (page 10, line 274).

  1. Some parts of the manuscript are too long in my opinion (especially at the end of the manuscript) and should therefore be shortened in order to keep the take-home message of this review clear, focused, and relevant for the readers. Otherwise, readers could be lost since a lot of information (even if they are of great interest) can be found in this work.

We thank you for valuing the importance of these complex topics that we have tried to condense as much as possible. We believe that you are referring to Chapter 4 (the longest of the review) and in fact we have used subheadings to help the reader navigate it. However, we have further tried to shorten this chapter by removing several specific examples from the text, which were included in the tables, and we hope this will improve the fluency. Nonetheless, if you still consider this chapter too long, we would be happy to reduce specific paragraphs if you could indicate which ones.

Reviewer 2 Report

The authors provided a well-documented overview of EVs and mitochondrial reprogramming mechanisms in cancer survival and progression. In addition to this, the review could represent a really interesting point of view in a field so dynamic and rich in potential future applications. The field of research focused on EVs is in continuous evolution and even if the article is well written, the introduction section could be improved by adding some recent works related to the need for new technologies able to associate a specific marker with an exosome subtype and this exosome subtype to a particular function and/or group of functions (PMID: 35141731 is just an example).

Figures look great and captions are clear and descriptive.

Nice work! 

I hope that my comments could be useful and I look forward to reading the revised version of the paper.

Good luck.

Author Response

Dear Reviewer 2,

Thank you very much for your time and effort in reviewing our manuscript. We are pleased to hear you found it well written, and interesting. We also appreciate your complimentary comments on our figures and captions as well as your encouragement. Please see below our point-by-point comments to your constructive suggestions:

The introduction section could be improved by adding some recent works related to the need for new technologies able to associate a specific marker with an exosome subtype and this exosome subtype to a particular function and/or group of functions (PMID: 35141731 is just an example).

Thank you for your comment, we have now added a couple of sentences discussing the need to develop new technologies allowing specific EV subtypes isolation (page 2 lines 68-71) and included the suggested reference in this new section (page 2 line 71).
